# Breaking Barriers: The Influence of Teachers' Attitudes on Inclusive Education for Students with Mild Learning Disabilities (MLDs)

Mahwish Kamran [1], Sohni Siddiqui [2,*] and Muhammad Shahnawaz Adil [3]

1 Department of Educational Sciences, Iqra University, Karachi 75300, Pakistan; mahwish.siddiqui@iqra.edu.pk
2 Department of Educational Psychology, Technische Universität Berlin, 10587 Berlin, Germany
3 Department of Management Sciences, Iqra University, Karachi 75500, Pakistan; adil.s@iuk.edu.pk
* Correspondence: s.zahid@campus.tu-berlin.de

**Abstract:** Inclusive educational practices demand social justice where all students with special educational needs have the same right to access education, irrespective of their special needs. Increasingly, across the world, teachers are supporting and defending the inclusion of students with disabilities in mainstream schools and classrooms. This is also the case in Pakistan, the site of this study. However, support and defense are not assurance that such policy is accepted by classroom teachers. Previous research studies have identified some barriers to inclusion and in this regard, the attitudes of teachers have been identified as a barrier worthy of consideration. The present research can contribute to developing insights by suggesting all the factors that can accommodate students with mild learning disabilities (MLDs). The research study aimed to determine the attitudes of teachers towards the education of students with mild learning disabilities, which are due to hereditary and neurobiological reasons that alter the functioning of the brain by affecting one or more intellectual processes associated with learning. The problems of processing can interfere with basic skills of learning, such as reading, writing, and mathematical skills. They can also interfere with higher-order skills, such as organizational skills, abstract reasoning skills, thinking skills, long or short-term memory, and attention span, in an inclusive classroom setting. Using a quantitative research method, data were gathered from N = 230 sample cases of teachers selected through a stratified sampling technique from 10 private primary inclusive schools and 10 private primary non-inclusive schools in Karachi. To identify the attitudes of teachers towards the inclusion of students with MLDs, teachers were asked to respond to validated and reliable tools used in different studies. The research applied variance-based structural equation modeling using the partial least-squares modeling method. Smart PLS 3.0 is software used for variance-based structural equation modeling, and the purpose of using it that the model involves mediation. This tool can show direct and indirect associations among variables simultaneously. The results revealed that there is a weak linear relationship between teachers' attitudes towards the education of students with a mild learning disability and their practices towards inclusive classroom settings. On the contrary, teachers' positive attitudes towards mild learning disabilities are predictors of inclusive classroom settings in schools. The weak linear association between teachers' attitudes and the provision of inclusive classroom settings showed that teachers are not prepared to accommodate mild learning disabilities. However, if teachers' self-efficacy is increased, then teachers are prepared to accommodate mild learning disabilities. It indicates that teachers with a greater sense of self-efficacy can play a significant role in creating an inclusive environment in schools by employing the provision of relevant resources. The present study recommended certain support mechanisms to school management and provided guiding principles to them on the specific resources required to meet the needs of students with a mild learning disability and to increase the self-efficacy of teachers.

**Keywords:** inclusive classroom settings; mild learning disability; self-efficacy; teachers' attitudes

## 1. Introduction

There has been an increasing concern among teachers about universalizing inclusive education since the early 1990s. Keeping in mind the Salamanca Statement and the Sustainable Development Goals (SDGs), which Pakistan endorses together with the Education for All declarations [1], researchers emphasize facilitating the diverse needs of students with disabilities so that they become a constructive part of conventional classrooms [2]. In line with the United Nation's Convention on the Rights of Persons with Disabilities (CRPD), education bodies are encouraged to integrate all students into mainstream learning environments and to address their diverse needs effectively in the classroom [3]. The CRPD was ratified by Pakistan in 2011 [4]. Additionally, it was asserted that all individuals, regardless of any disabilities, should enjoy basic human rights in line with their peers, such as access to basic education. It is also important to highlight that basic education is a non-progressive right, which means that it should immediately be made available by Governments who ratified the CRPD—governments cannot wait until budgets become available to realize basic education. This is particularly important in low- and middle-income countries. The CRPD also recognizes the need for adaptation to facilitate people with disabilities so that their natural rights are better protected against exploitation. According to the context of this research, though Pakistan supports and defends the presence of students with disabilities in mainstream schools and conventional classrooms, there is a substantial gap in planning and preparation [5,6]. Teachers' insufficient knowledge, skills, attitudes, and financial constraints result in hesitance towards inclusive practices [7,8], and they require external support to promote successful inclusion [9]. Therefore, it is worthwhile to conduct research that can help in the successful implementation of inclusion.

Between 1999 and 2010, Pakistan's education expenditure decreased from 2.6% to 2.3% of its gross national product (GNP), according to the Education for All Global Monitoring Report. Education receives only 9% of government spending in Pakistan, ranking the country 113th out of 120 countries on the Education Development Index [10]. Pakistan has the second-highest number of out-of-school children with 22.84 million children [11]. Despite being a signatory to international policies promoting literacy rates and universal primary education, Pakistan has failed to effectively implement these policies. Various factors, including geopolitical instability, national instability, inaccessibility, underfunding of education, and economic pressures, contribute to the complex problems in the education system [4]. While urban areas generally have better education systems and increasing literacy rates, there are still significant challenges, leading to the partial or complete exclusion of many students from education [9]. Pakistan's education system is divided into different grades, ranging from pre-primary to higher secondary schools. The primary school age group starts at 5 years old. Pakistan's schools are state-run (public) or private. Currently, there are 76,674 privately run institutions compared to 194,151 state-run institutions [10]. The public sector serves 26.63 million students, while the private sector caters for 13.96 million out of approximately 40 million [10]. Pakistan faces challenges in inclusion, with very few inclusive schools and limited enrollment of students with disabilities at the appropriate age for primary education. Early intervention for students with disabilities is a challenge. However, the National Education Policy of 2017 provides specific guidelines for implementing inclusive education in Pakistan [11].

Typically, B.Ed. programs primarily cover general education theories, while M.Ed. (Master of Education) programs introduce specialized courses. However, some universities offer bachelor's programs in Special Education, specifically designed to train teachers interested in working with students with disabilities. The ratio of teachers to students is not universally standardized and varies significantly between institutes.

Research studies have shown an association between teachers' attitudes and disability type or severity. For instance, teachers are more likely to be comfortable accommodating students with physical and sensory disabilities, for instance, compared to those with cognitive and learning disorders [2]. Moreover, in the context of the developing country, a research study conducted by Donohue and Bornman (2015) highlighted that the provision

of resources and training can develop an attitude of inclusion among teachers [12]. Similar to the current research, a study was conducted by Bornman and Donohue (2013) whose findings state that training of teachers can pave the way toward inclusion [13]. The current research study aims to determine teachers' attitudes toward accommodating students with mild learning disabilities in schools in Pakistan. It also suggests factors that can promote inclusion in schools and turn inclusion into a reality.

*1.1. Definitions of Key Terms*

1.1.1. Inclusive Classroom

A classroom where students with mild learning disabilities and students without disabilities are facilitated in the same space.

1.1.2. Positive Attitudes towards Mild Learning Disability

This refers to the accommodation of students with disabilities in a mainstream setup.

1.1.3. Teachers' Sense of Self-Efficacy

This refers to teachers' confidence in their abilities to bring about the intended results of students' engagement and learning. Self-efficacy is an individual type of confidence in which one can accomplish objectives suitably and effectively.

1.1.4. Mild Learning Disability

A child who is not accompanied by a mild learning disability is generally capable of holding a conversation and communicating most of their needs and wishes. However, these students might require some support to comprehend abstract concepts.

**2. Literature Review**

*2.1. Inclusive Settings in Schools*

The basic principle of inclusion proclaims that all individuals are distinctive in various ways and should be understood and accommodated according to their uniqueness and differences. It proposes such a setting where all students are welcome to gain access to all educational opportunities, irrespective of any learning incapacity or physical disorder [8]. Inclusive education is all about the pursuit of justice, contribution, and insight into social responsibility. It emphasizes the omission of obstacles of segregation and repression, while focusing on the happiness of all students, including disabled students [1]. Hence, inclusion is founded on an affirming conceptualization of difference where learner diversity is regarded as a positive resource and something to be celebrated. Importance is given to the quest for variation, with great emphasis on the significance of teaching how to live with one another and how to be aware of our common humanity [14].

Adam Smith's equity theory emphasizes the importance of finding whether the distribution of resources is impartial and equal among all relational partners. Inclusion is an approach that is beyond the notion that all students should be educated in the same place. It emphasizes impartiality and the positive participation of every individual [15]. Inclusive education aims to include all the students in a regular classroom setting in mainstream schools where students with and without disabilities are accommodated based on the principles of societal impartiality and civil rights (Individuals with Disabilities Education Act) [7,16,17]. An inclusive approach to the education of students with learning disabilities has often been regarded as the only way to maximize opportunities to achieve and participate in education and society [18]. Researchers have identified different studies and stated that to become an inclusive school, one needs to find and remove barriers that hinder inclusion [19]. It involves teachers recognizing how they can support students with mild disabilities. In Pakistan, the current state of inclusion is unwelcoming, especially in public schools where inclusion is nonexistent [5]. In contrast, in the private sector, inclusion is taking place, but often with hefty fees attached [20].

### 2.2. Teachers' Attitudes towards Mild Learning Disabilities (MLDs)

It is pertinent to understand how mild learning disability is conceptualized in Pakistan. In the Pakistani context, it is defined as a broad term that covers a group of possible causes, signs and symptoms, treatments, management, outcomes, and effects on the life of any mild neural disorder. Learning disabilities have different manifestations in different individuals and not all students with learning disabilities need to display all the characteristic features related to them [21]. Learning disability is a cognitive disability that affects how an individual comprehends information and responds to it. They might encounter difficulties comprehending intricate or convoluted knowledge, grasping new knowledge, and handling it autonomously. It is of great interest as teachers are usually reluctant to include them in mainstream classroom settings [22], and it is worthwhile studying the reasons for this reluctance. Consequently, the present study focuses on students with mild learning disabilities. The acquisition and utilization of fundamental language skills, such as listening, speaking, reading, and writing, can be impeded by learning disabilities. These disabilities may affect various aspects, such as phonemic awareness, word recognition, intellectual capacity, spelling, written communication, and mathematical abilities, such as calculation and problem solving. Additionally, learning disabilities may also impact organizational skills, social awareness, social communication, and adjustment. Experts classify learning disabilities into four types: mild, moderate, severe, and profound. Previous research suggests that IQ scores may carry less significance compared to the extent and nature of intervention needed for individuals with learning disabilities, as noted by Smith [23].

Inclusion emphasizes the revision of widely practiced approaches to teaching and learning. This requires curriculum modification, which should be the school's responsibility. According to established theory in the field of inclusive education, there are robust statistics available that all students grasp differently, for instance, by close observation or by impersonating others; explicitly, students with cognitive disorders learn speedily among their classmates, meaning these students learn more rapidly when they are in inclusive classrooms than similar students who are not in inclusive classrooms [24]. This results in the creation of a collective teaching and learning environment through which a multiplicity of students can be promoted by encouraging an empathetic attitude [24].

### 2.3. Teachers' Sense of Self-Efficacy (TSES) Facilitates Inclusive Settings in Schools

The practical implementation of inclusion in mainstream schools can be achieved when teachers and principals are willing to embrace its philosophy and unique requirements. Previous studies, including those by Kazmi et al. [8] and Kamran et al. [9] in Pakistan, have reported that teachers often have varying attitudes towards inclusive education. Additionally, teachers commonly express practical concerns that may shape their attitudes towards inclusion. These concerns may include managing individualized time demands of students with disabilities while not disrupting other students in the classroom, addressing teacher hesitance through training on inclusive practices for students with disabilities, ensuring adequate provision of resources, and enhancing competency in supporting inclusive practices, as highlighted by Gaines and Barnes [25].

Researchers have stated that teachers need to be trained to meet inclusive education demands [1]. Training can make them confident in their abilities with improved knowledge, skills, and attitudes. As a result, engaging students, developing instructional strategies, and managing classrooms can be enhanced. Furthermore, a research study indicates that teachers' attitudes towards inclusive education improved through professional development [7]. It was also suggested that workshops should be organized for the benefit of teachers working with students with disabilities [6,26]. Another research study carried out in the Pakistani context revealed that trained teachers were more efficacious towards including students with disabilities [27]. The research carried out so far in the field of teacher education with special reference to students with disabilities claims that without a professionally developed teacher, inclusion can never become a reality [28].

Bandura introduced the concept of self-efficacy three decades ago as a critical factor in human motivation. Self-efficacy refers to an individual's perception of their ability to achieve desired levels of performance, which significantly influences their actions and outcomes in life [29]. According to Bandura's social cognitive theory from 1986, an individual's self-referential beliefs act as a mediator between knowledge and actions [30]. Bandura considers self-reflection as an intrinsic human capability, through which individuals evaluate and modify their attitudes, which can play a crucial role in promoting inclusion and addressing diversity. Tschannen-Moran and Woolfolk Hoy (2001) described self-efficacy as a teacher's confidence in their ability to facilitate effective student performance [31]. Self-efficacy in instructional approaches, student's engagement, and classroom management is crucial for successful teaching and learning. In the literature, self-efficacy belief is highlighted as a significant mediator for attitude and attitude change [32,33]. Therefore, a teacher's sense of self-efficacy can greatly contribute to successful inclusive practices.

There are limited research studies that can directly compare teachers' attitudes toward including students with MLDs; the results of this research study contribute to special education research. Educational experts must examine the results and determine policies to improve regular education instructors' willingness to have students with a mild learning disability in their classroom. This notion clarifies that more preparation is required to facilitate disabilities as it increases self-efficacy. Especially for students with moderate learning disabilities, school management should provide opportunities for continuous professional development. Teachers need to be better informed about the most effective teaching practices to promote their educational, societal, and behavioral abilities [34].

## 3. Research Methodology

### 3.1. Research Design

The objective of the research study was to investigate the perspectives of teachers in Pakistan regarding the inclusion of students with mild learning disabilities in a mainstream classroom, utilizing a survey method and quantitative research approach. After obtaining permission from the authors and under the supervision of a supervisor, a comprehensive questionnaire was adopted. It was a previously published questionnaire; therefore, it was reliable and valid. The tool helps to collect data to determine the attitudes of teachers in the context of Pakistan. The survey form, along with a letter of consent, was initially emailed to schools, and upon gaining approval, it was distributed to principals and coordinators. A pilot study was conducted following the recommendations of Saunders et al. (2009), by administering questionnaires to 10 teachers [35]. This small-scale pilot study aided the researchers in refining their research questions. Hypothesis testing was conducted using Smart PLS 3.0 and SPSS 22.

### 3.2. Research Questions

This research study addressed the following research questions:

RQ 1: What are primary school teachers' attitudes towards mild learning disabilities?

RQ 2: Is there a mediating effect of teacher self-efficacy on the relationship between teacher attitudes and inclusive classroom settings, such that this relationship is stronger when self-efficacy levels are higher?

### 3.3. Research Instrument

Part A: The first section of the research instrument was comprised of demographic questions that gathered information on gender, age, qualification, and type of initial teacher program.

Part B: The second section of the research instrument was adapted from Avramidis et al.'s (2000) study to measure teachers' attitudes [36]. The selected instrument was based on the three-component model of attitude and was considered appropriate for measuring attitudes towards inclusion. The attitudes were assessed in terms of belief (cognitive component, 6 items, 6-point Likert Scale: 1 = strongly disagree, 6 = strongly agree;

example statement: "I am actively developing my skills to teach students with mild learning disabilities in my classroom"); intentions (conative component, 6 items, 6-point Likert Scale: 1 = strongly disagree, 6 = strongly agree; example statement: "I am willing to develop my skills to teach students with mild learning disabilities in my classroom"); and emotions (affective component; 7 items, 6-point Likert Scale: 1 = strongly disagree, 6 = strongly agree; example statement: "If a new student described as having mild learning disability and behavioral disorder was about to join your class tomorrow, how would you feel?"). Both cognitive and conative components were used to assess teachers' intentions towards inclusion, with questions formulated to gauge readiness towards inclusion for teachers from schools that were not imparting inclusive education (e.g., "I will accommodate students with mild learning disabilities in my classroom"), while statements from teachers already working in inclusive settings expressed actual accommodations made (e.g., "I have accommodated students with mild learning disabilities in a classroom").

The scale used to measure the affective component employed a semantic differential scale with seven items consisting of bipolar adjectives, such as "disinterested-interested"; "negative-positive"; "uncomfortable-comfortable"; "unconfident-confident"; "worried-self-assured"; "unhappy-happy"; and "pessimistic-optimistic". It should be noted that only the second component, i.e., intentions, conveyed and described attitudes, while the belief and emotions components did not directly reflect attitudes. The conative component assessed an individual's inclination to act.

Part C: The third part of the questionnaire was adopted from Tschannen-Moran and Woolfolk Hoy's (2001) study to measure teacher self-efficacy in order to gain insights into the factors that pose challenges for teachers when instructing students with learning disabilities [31]. The survey questionnaire consisted of 11 items (example statement: "*How much do you believe you can do to get* students *to follow classroom rules?*") with a 5-point Likert scale (1 = Nothing, 5 = A great deal) that assessed teachers' efficacy in engaging students, developing instructional strategies, and managing classrooms.

### 3.4. Sampling Technique

This research study involved teachers from private primary schools, both inclusive and non-inclusive, in Karachi, Pakistan. The primary schools were selected as there was a research gap and very few research studies were conducted to find teachers' attitudes toward inclusive education in Pakistan [4]. The inclusive schools were identified through a non-profit organization, which provided the researcher with a list of such schools. The research data were collected using stratified sampling, wherein the entire population was divided into strata, namely 10 private inclusive schools and 10 private schools, which did not include students with disabilities. Random selection was performed from each stratum. This approach ensures that the sample population is representative of the entire population, following the methodology proposed by Creswell [37]. The data were collected from 237 teachers, but only the final 230 teachers' data were analyzed after removing 7 univariate and multivariate outliers using Smart PLS (partial least-squares method) 3.0 to support the mediation model of the study.

### 3.5. Ethical Considerations

Each participant in the research study was provided with a consent form, and it was made clear to them that they had the right to withdraw from the study at any time without any bias. Additionally, the potential participants were given an information sheet that thoroughly explained the nature of the research study. Those who signed the consent form were verbally reassured that their identity would remain confidential and would not be disclosed to anyone. To ensure privacy and confidentiality, it was clarified that when reporting the research findings, pseudonyms would be used for schools, teachers, principals, and students to prevent their identification.

## 4. Analysis of Results

### 4.1. Respondent Profile

For data analysis, this study utilized 230 valid cases (115 from inclusive settings and 115 from non-inclusive setups) after removing 7 cases as part of the data screening process. Out of the 230 respondents, 208 (90.4%) were female teachers and 22 (9.6%) were male teachers. In terms of age, 73 (31.7%) fell within the age group of 18–25 years, 94 (40.9%) were in the age bracket of 26–35 years, 49 (21.3%) were between 36–45 years old, while 8 (3.5%) and 6 (2.6%) were in the age groups of 46–55 and above 55 years, respectively. The majority of teachers had a graduate degree (N = 116, 50.4%) or a master's degree (N = 94, 40.9%), while 10 (4.3%) had completed a college degree, 8 (3.5%) had a higher diploma, and 2 (0.9%) held a doctorate degree.

### 4.2. Factor Analysis of the Instrument

The empirical method in this study employed the PLS (partial least squares) technique to calculate results from the sample (refer to Table 1). The outer model was used to assess the association between the latent variable and Likert scale items. The latent model was created with accurate calculations to enhance the overall model, following the approach proposed by Fornell and Larcker [38].

**Table 1.** Factor Loadings and Convergent Validity.

| Construct | Items | Factor Loadings | Composite Reliability | Average Variance Extracted |
|---|---|---|---|---|
| Cognitive component | CC1 | 0.749 | | |
| | CC2 | 0.856 | | |
| | CC3 | 0.788 | 0.959 | 0.659 |
| | CC4 | 0.825 | | |
| | CC5 | 0.845 | | |
| | CC7 | 0.763 | | |
| Conative component | CC1 | 0.745 | | |
| | CC2 | 0.831 | | |
| | CC3 | 0.884 | 0.959 | 0.659 |
| | CC4 | 0.854 | | |
| | CC5 | 0.820 | | |
| | CC7 | 0.768 | | |
| Affective component | AC1 | 0.834 | | |
| | AC2 | 0.855 | | |
| | AC3 | 0.870 | | |
| | AC4 | 0.891 | 0.954 | 0.747 |
| | AC5 | 0.849 | | |
| | AC6 | 0.884 | | |
| | AC7 | 0.869 | | |
| | TSES1 | 0.650 | | |
| Teachers' | TSES2 | 0.742 | | |
| sense of | TSES3 | 0.735 | | |
| self- | TSES4 | 0.677 | | |
| efficacy (TSES) | TSES5 | 0.665 | | |
| | TSES6 | 0.733 | | |
| | TSES7 | 0.685 | 0.913 | 0.5 |
| | TSES8 | 0.688 | | |
| | TSES9 | 0.693 | | |
| | TSES10 | 0.731 | | |
| | TSES11 | 0.691 | | |

### 4.3. Reliability and Convergent Validity

The reliability of an instrument is commonly evaluated using measures such as composite reliability, with a threshold value typically set at 0.6 or higher [39]. Similarly, convergent validity is assessed using average variance extracted (AVE), with a recommended threshold value of 0.50 or higher, and factor loading values above 0.6, indicating that convergent validity was established [40] (refer to Table 1).

### 4.4. Discriminant Validity

Construct validity can also be confirmed by discriminant validity, which can be assessed by the Fornell and Larcker (1981) discriminant validity criteria [38]. In Table 2, it can be observed that the diagonal values are superior to the other values in the corresponding column and row, providing evidence for the discriminant validity of the outer model.

**Table 2.** Fornell–Larcker Criterion.

|  | CC | AC | TSES |
|---|---|---|---|
| Cognitive and conative component (CC) | 0.812 | | |
| Affective component (AC) | 0.391 | 0.864 | |
| Teachers' sense of self-efficacy (TSES) | 0.442 | 0.468 | 0.700 |

### 4.5. Model Fitness

The precision of the inner model can be assessed using the coefficient of determination, $R^2$. As explained by Hair et al. (2012), $R^2$ reflects the cumulative impact of independent (exogenous) variables on dependent (endogenous) variables [40]. A value of 0.25, 0.50, or 0.75 is typically considered indicative of weak, moderate, or substantial levels of predictive accuracy, respectively [40] (refer to Table 3).

**Table 3.** $R^2$ and $R^2$ Adjusted.

|  | $R^2$ | $R^2$ Adjusted |
|---|---|---|
| Cognitive and conative component (CC) | 0.565 | 0.555 |
| Teacher's sense of self-efficacy (TSES) | 0.301 | 0.289 |

### 4.6. Inner Model Testing

In this study, the hypothesized model relations were tested using standard path coefficients in Smart PLS 3.0 software. It is important to note that conventional *t*-tests are not calculated in Smart PLS, as highlighted by Barclay et al. [41]. Instead, to estimate the accuracy of the results, bootstrapping, as suggested by Chin [42], is applied. A bootstrap sample of 5000 was used to analyze the significance of the path coefficients in this study (refer to Table 4).

**Table 4.** Inner Model Results.

| Hypothesis | Path | Original Sample | T-Value | *p*-Value | Decision |
|---|---|---|---|---|---|
| There is an association between teacher's attitudes towards mild learning disabilities (AC) and inclusive settings in schools (CC) | AC→CC | −0.101 | 0.827 | 0.408 | Rejected |
| Teacher's sense of self-efficacy (TSES) mediates the relationship between mild learning disabilities (AC) and inclusive settings in schools (CC). | TSES→CC | 0.136 | 2.105 | 0.035 | Accepted |

*4.7. Hypothesis Testing and Path Coefficient for Direct and Indirect Hypothesis*

To generate the path coefficient, a PLS algorithm was run. After that, researchers continued running bootstrapping with 5000 bootstrap samples to meet the conditions suggested by [40] to generate *p*-values. If the *p*-value is less than 0.05, an alternative hypothesis is accepted and if the *p*-value is greater than 0.05, a null hypothesis is accepted. To assess the first hypothesis, a direct relationship was established between teachers' attitudes towards mild learning disabilities and inclusive settings in schools and found a statistically insignificant relationship between the two ($p > 0.05$). Hence, the hypothesis was rejected. To assess the second hypothesis, Figure 1 shows that the indirect relationship was established as teachers' sense of self-efficacy (TSES) mediated a relationship between mild learning disabilities (AC) and an inclusive setting (CC). However, it can be observed that c' or a*b had a significant *p*-value ($p < 0.05$), whereas path c had an insignificant *p*-value, hence the relation was indirect-only mediation [43]. In this case, the hypothesis was accepted.

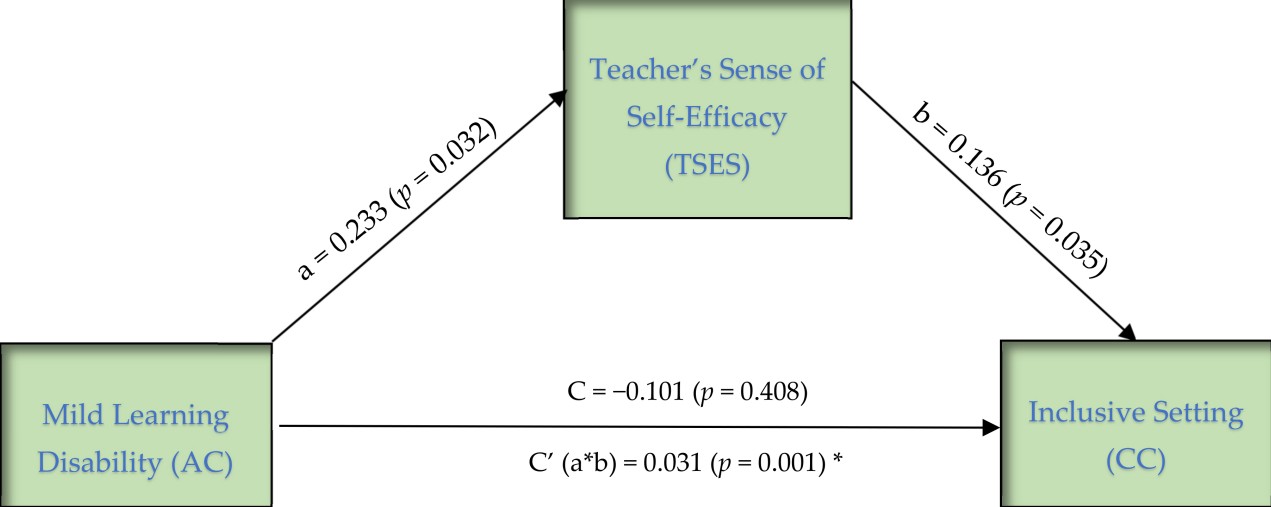

**Figure 1.** Teachers' sense of self-efficacy mediates the relationship between mild learning disabilities and inclusive setting. * *p*-value is statistically significant at the 1% level of confidence ($p < 0.01$).

## 5. Findings and Discussion

The objective of the present research study was to investigate teachers' attitudes towards accommodating students with mild learning disabilities in schools in Pakistan. Additionally, the study proposed that factors such as teachers' self-efficacy can facilitate inclusion in schools and contribute to making inclusion a tangible reality.

The implementation of strategies can support the accommodation of students with mild learning disabilities in regular classrooms, thereby making inclusion a reality [6]. Teachers expressed satisfaction in including students with mild learning disabilities by utilizing available resources to support their management in a conventional classroom [8]. Research indicated that teachers' belief in their ability to accommodate students with learning disabilities positively impacted their performance, with improved performance observed when teachers have poise in their ability to effectively accommodate these students.

The first research question of the study was to determine the impact of primary school teachers' attitudes towards mild learning disabilities. The findings show that teachers' attitude towards mild learning disabilities is insignificantly related to inclusive settings in schools. According to the previous study, teachers have an adverse attitude towards learning disabilities if they are less severe, which is in agreement with previous research studies [5,22]. In the current research study, teachers were usually found to be more inclined towards including students with, for instance, physical and sensory disabilities compared to those with cognitive and learning disorders [17]. Earlier research studies also supported the finding that teachers have negative attitudes towards mild learning

disabilities in comparison with sensory, physical, or behavioral disorders [19]. The reason for this reported by other researchers is that usually students with learning disabilities have multiple disabilities, therefore making them difficult to handle, and this is consistent in Pakistan [5]. The results are consistent with the current study, which is an indicator that there are some types of inadequacies regarding inclusive education. These inadequacies include reluctance to include students with disabilities, lack of willingness, and above all, a lack of acceptability of students with disabilities [8]. In 2010, Uzair-ul-Hassan et al. identified several key factors contributing to teachers' reservations about inclusive education [20]. One significant issue is a lack of skills in managing large classes with diverse needs. While teachers generally accept the presence of students with disabilities in their regular classes, they often doubt the effectiveness of inclusion, especially when faced with a high student-to-teacher ratio. Other reasons for negative attitudes towards inclusion include inadequate resources and training to support students with special needs, increased workload and stress, inappropriate curricula, uncooperative school administrators, and unsupportive parents. Furthermore, the absence of special education teachers in regular schools poses an additional challenge for inclusive education.

The second research question examined whether teachers' self-efficacy served as a mediator between their attitudes and inclusive setups in schools, with a stronger association observed when self-efficacy is higher. The results indicate that teachers' self-efficacy indeed mediates this effect. Previous research studies have shown that a dearth of self-efficacy regarding the accommodation of students with mild learning disabilities is linked with undesirable attitudes towards including students with disabilities. Hence, teachers' sense of self-efficacy plays a significant mediating role between attitudes and inclusive settings. This finding aligns with previous research that highlights how increased self-confidence among teachers enables them to effectively accommodate students with disabilities [5,6]. It is possible that teachers' attitudes are linked to their efforts in finding solutions to challenges related to the accessibility of human, physical, and societal support, as well as the requisite knowledge, skills, and positive attitudes needed to effectively accommodate students with disabilities. In this regard, increasing self-efficacy plays a pivotal role in developing attitudes towards inclusion and in turn towards paving the path for promoting inclusion. Knowledge, skills, and positive attitudes seem to be significant elements that impact teachers' aptitude to modify teaching practices.

The knowledge area of teachers can be categorized into subject matter knowledge, pedagogical knowledge that comprises strategies related to classroom management and instructional techniques, and pedagogical content knowledge that is extremely important to delivering specific subject matter to each student in a defined context. Emphasizing the significance of teachers' knowledge is crucial, with particular attention needed to enhance and improve their pedagogical content knowledge related to students with mild learning disabilities to positively impact their attitudes towards inclusive schools. Additionally, research indicates that information about mild learning disabilities has a positive influence on attitudes [4]. This suggests that general plans for inclusive teaching may not be sufficient and that developing insight into specific diagnoses and how they can affect students with mild learning disabilities and their learning can improve pedagogical content knowledge. The present research contributes to this insight by identifying factors that can accommodate students with disabilities.

Furthermore, teachers' self-efficacy belief is determined by three factors: efficacy in engaging students, efficacy in instructional strategies, and efficacy in classroom management [31]. Teachers with higher self-efficacy in addressing all three factors are more likely to mediate the relationship between their attitudes and inclusive setups in schools, as the current research results clearly show that teachers with higher self-efficacy are more inclined towards inclusion and accommodating mild learning disabilities. Therefore, it is recommended that the mediating role of self-efficacy be considered, and organizations should focus on providing the necessary resources to enhance self-efficacy, as successful inclusion may not be possible without it.

## 6. Conclusions

The results of the research illustrated that more struggles need to be overcome to teach students with disabilities in Pakistan. Generally, teachers embrace an affirmative outlook toward the inclusion of students with disabilities. However, a partnership between mainstream schoolteachers and special education teachers is imperative. In this regard, training workshops should be conducted to facilitate teaching students with disabilities. It is also important for schools to ensure the provision of adequate resources. Inclusion entails support by all stakeholders, including school management, principals, parents, teachers, and peers.

The research study revealed that teachers who have high self-efficacy tend to be more assertive in accommodating the instructional needs of students with mild learning disabilities. However, they may find it challenging to manage students with moderate to severe learning disabilities in conventional classrooms. This less supportive attitude towards students with moderate to severe disabilities can be attributed to the limited capacity of teachers and the lack of available resources. The literature indicates that teachers who are not professionally developed may feel reluctant to accommodate students with disabilities [4], and the unavailability of resources, such as assistive devices, also plays a crucial role in influencing teachers' reluctance [2].

This research study addressed teachers' attitudes toward inclusion and how those attitudes are revealed in their actions in the classroom. A teacher's attitude towards inclusive arrangements does affect the success of their inclusive classroom setup. For that reason, further investigation should examine elements that can increase teachers' self-efficacy so that teachers demonstrate positive attitudes to facilitate moderate to severe learning disabilities and pave the way towards complete inclusion.

## 7. Recommendation

It is essential that teachers of students with disabilities must have adequate knowledge, skills, and attitudes so that they can become demonstratively prepared to provide demanding and individualized instructions to their students. Consequently, it is the responsibility of school management to provide resources that should be accessible to all stakeholders in the school to support teachers accommodate students with disabilities. Considering the limitations of the present study, a strong prospect is to gather inductive and longitudinal data on the improvement and change in teachers' attitudes towards inclusion.

Future research requirements are to explore in detail some of the deeper matters concerning inclusive education as inclusion is the only way to discourage exclusion. It is imperative to make general education teachers more confident to accommodate students with mild learning disabilities, to execute students' personalized instructional strategies, and to cooperate to implement required adjustments. If these measures are taken, it is possible that general education teachers' attitude towards including students with disabilities will be enhanced.

## 8. Limitations

Given the limitations of the current study, there is potential to collect inductive and longitudinal data to examine the changes and improvements in teachers' attitudes towards inclusion. It should be noted that the research study was limited to private sector schools, encompassing both inclusive and non-inclusive settings, while public sector schools were not included in the analysis.

**Author Contributions:** Conceptualization, M.K; methodology, M.K. and S.S.; validation, S.S.; formal analysis, M.K. and M.S.A.; investigation, M.K.; resources, M.K., M.S.A. and S.S.; data curation, M.K.; writing—original draft preparation, M.K; writing—review and editing, S.S.; supervision, M.K.; project administration, M.K.; funding acquisition, S.S. All authors have read and agreed to the published version of the manuscript.

**Funding:** The researchers acknowledge the support from the German Research Foundation and the Open Access Publication Fund of TU Berlin.

**Institutional Review Board Statement:** The researchers followed basic ethical principles and the APA's ethical code. Ethical review and approval were not required for the study on human participants in accordance with the local legislation and institutional requirements. However, the entire study and questionnaire were designed and reviewed under the supervision of the first author's research team consisting of educational psychologists and educationalists from a private university in the Metropolis City of Pakistan who are well acquainted with Pakistan's educational system. The reviewers found no potential conflict of interest or harm to participants, nor any activities that transcended beyond the ethical code of conduct. The researchers designed the research to ensure integrity, quality, and transparency. For this purpose, the research participants were informed about the purpose, method, and intended uses of the research. The researchers also explained to the participants about what their participation in the research entails and how it could contribute to academia. It was additionally ensured that their confidentiality and anonymity were respected. They were also informed that their participation was voluntary and have the right to withdraw from the study at any time. Their dignity and autonomy are protected and respected at all times. They were comforted that risk or harm would be avoided in all instances. In the current research study, gatekeepers were required to cooperate and consent to access to research participants. To inform the participants about the research objectives, processes, and outcomes, a letter of information was shared. The consent form was also handed over to the gatekeepers of the respective institutions. Research participants were required to consent via signature on the form. They were also informed about secure data storage both during and after the project. Access to the data was limited to researchers and supervisors only, and measures were taken to ensure privacy and confidentiality. The use of pseudonyms was clarified for schools, teachers, head teachers, and students to prevent identification during the reporting of research findings. It's worth noting that there were no potential conflicts of interest.

**Informed Consent Statement:** Informed consent was obtained from all participants involved in the study.

**Data Availability Statement:** The raw data supporting the conclusions of this article will be made available by the authors upon request, without undue reservation.

**Conflicts of Interest:** The authors declare no conflict of interest.

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
