# Peer review of "Breaking Barriers: The Influence of Teachers’ Attitudes on Inclusive Education for Students with Mild Learning Disabilities (MLDs)"

_education, doi:10.3390/educsci13060606_

Round 1

Reviewer 1 Report

A compelling contribution to research on teachers' attitudes toward students with learning disabilities. The finding that teachers in mainstream schools tend to focus around students with more prominent and severe disabilities due to limited resources is certainly not new. Neither is the insight that teachers' engagement can be increased by promoting their self-efficacy expectations. Nevertheless, this connection and especially the support needs of teachers in mainstream schools cannot be emphasized often enough. This article does this and provides the corresponding empirical evidence in a methodologically clean, transparent, and convincing manner. A textbook manuscript.

Author Response

Thank you, reviewer, for appreciating this work.

Reviewer 2 Report

See my attached sheet. 

Author Response

Respected reviewer. Thank you for your valuable comments. Suggested changes are made and highlighted as red. Please refer to the attached file.

Reviewer 3 Report

To be shared by Editor as seen fit.

Please see extensive comments to editor.

Author Response

Respected reviewer. Thank you for your valuable comments. Suggested changes are made and highlighted as green. Please refer to the attached file.

Round 2

Reviewer 3 Report

Thank you for the opportunity to re-review this paper. From the extensive “track changes” and comments made in the document it is evident that the authors had carefully considered original comments.  I noted that many of my earlier comments had been addressed – thank you.

Some minor corrections are still needed:

·       Line 34:  Please change  wording “Teachers with a greater sense…”

·       Line 48: Please adapt” “In line with the United Nation’s Convention on the rights of Persons with Disabilities (CRPD), education bodies….”

·       Line 56, please change “The UN Convention” to “The CRPD…”

·       In line 46 where the SDGs are mentioned – link this to line 64  to complete the consent and not speak about it in a fragmented way, e.g.  In line 46 say “…and the Sustainable Development Goals (SDGs), which Pakistan endorses together with the Education for All declarations [9], researchers emphasize….”

·       Thank you for adding lines 68 to 95, it really helps to understand the education landscape in Pakistan.

·       Line 102 – not sure what the “similar” refers to – similar to the current study or similar to the previous Donohue & Bornman study?

Line 103 reads strange – it seems as if a word is missing? “and findings state…?

·       Line 109, 146, 200, 201, 378, 393,397, 412, 461 – please change “children” to “students” in order to ensure consistency throughout the manuscript. (Please search for these terms and correct throughout – I did not point them all out).

·       Line 114 and 124, 139, 188, 213, please change “learners ” to “students” in order to ensure consistency throughout the manuscript. (Please search for these terms and correct throughout – I did not point them all out).

·       Line 202 - please change “educator ” to “teacher” in order to ensure consistency throughout the manuscript

·       Line 230 ‘ please add “perspectives of teachers in Pakistan regarding…”

·       Line 356 please add a hyphen between p-values

·       Line 461, please change “… reluctant to accommodate students with…” (as accommodate is more in line with the terminology of inclusion and the CRPD.

This study would benefit from language editing due to some awkward phrasing, language inconsistencies, and language and punctuation errors. I tried to point some of these out - but I am not a English first language speaker.

Author Response

Revisions are addressed and response to each comment is mentioned in the attached file.
